# Alcohol and Drug Consumption among Drivers before and during the COVID-19 Pandemic: An Observational Study

Maricla Marrone [1], Fortunato Pititto [1,*], Alessandra Stellacci [1], Simona Nicolì [1], Luigi Buongiorno [1], Benedetta Pia De Luca [1], Lucia Aventaggiato [2], Giuseppe Strisciullo [2], Biagio Solarino [1] and Marcello Benevento [1]

1 Interdisciplinary Department of Medicine, Section of Legal Medicine, University of Bari "Aldo Moro", Piazza Giulio Cesare 11, 70124 Bari, Italy
2 Forensic Toxicology Laboratory, Interdisciplinary Department of Medicine, University of Bari "Aldo Moro", 70124 Bari, Italy
* Correspondence: fortunato.pititto@uniba.it

**Abstract:** Restrictions imposed during the COVID-19 pandemic might have changed recreational habits. In this study, the results of toxicological tests for alcohol and drugs in blood were compared among drivers stopped at roadside checks in the periods before (1 January 2018 to 8 March 2020) and after the lockdown measures (9 March 2020 to 31 December 2021). A total of 123 (20.7%) subjects had a blood alcohol level above the legal limit for driving of 0.5 g/L, 21 (3.9%) subjects tested positive for cocaine, and 29 (5.4%) subjects positive for cannabis. In the COVID-19 period, the mean blood alcohol level was significantly higher than in the previous period. Cannabis use, which was more frequent among younger subjects, was statistically associated with cocaine use. There has also been a quantitative increase in alcohol levels in the population with values above the legal limits, indicative of greater use of alcohol in the population predisposed to its intake.

**Keywords:** alcohol; mental health; drug; COVID-19; lockdown; driver; cannabis; cocaine

## 1. Introduction

On 9 March 2020, the World Health Organization (WHO) declared a pandemic status, following the unexpected and unpredictable spread of a predominantly respiratory infectious disease, SARS-CoV-2 disease.

The pandemic caused an upheaval in the daily lives of millions of people [1]; the necessary containment measures, including mainly physical distancing and isolation, had harmful consequences on the physical and mental health of the world population [2–5].

In order to reduce the spread of the virus, national and international institutions almost everywhere in the world set up tools such as case isolation, quarantine of contacts, and physical distancing in moments of sociality. Psychological reactions to quarantine, such as frustration, loneliness, and worries about the future, were the most common and represented well-known risk factors for several mental disorders, including anxiety, affective disorders, and post-traumatic stress disorder [6–8].

Italy was the first European country to apply contagion containment measures; on 10 March 2020, the state of "lockdown" officially entered into force, imposing restrictions on free movement [9]. The lockdown imposed the suspension of social and cultural activities (theaters, cinemas, gyms, sports centers), partial closure of other economic and industrial activities, and prohibited gatherings of people in public places or places open to the public. In May 2020, the government imposed even stricter rules, with the closure of all non-strategic economic activities, including schools and universities. Hence, the lockdown period caused major changes in people's habits [10].

In Italy, road traffic underwent a significant reduction of up to 80% in the period March–April 2020 [11]. In fact, movements were only allowed for specific reasons and

when documented by specific certifications. As a consequence, in 2020, there was a decrease (−31.3% compared to the previous year) in road accidents [12]. The reduction of circulating cars also produced a reduction in fuel consumption in many EU states during the pandemic [13].

During the lockdown, alcohol was the most used substance, even on a daily basis, in order to reduce feelings of anxiety and/or stress, sleep better, and escape from reality [14].

Several studies in fact highlighted an increase in the tendency to replace drugs/illicit substances with others that are potentially dangerous, but more easily available, such as alcohol and benzodiazepines [15]. In France and Belgium, the closure of bars and restaurants during the lockdown resulted in an overall reduction in alcohol consumption, especially among young adults. On the other hand, adult people between the age of 35 and 50 years reported that they drank more during the lockdown, even though they could not go to the aforementioned commercial services [16]. Murthy et al. stated that the pandemic played a significant role in increased alcohol use [17]. Alladio et al. specifically found a general reduction in alcohol consumption (probably for the same reasons previously identified) during the lockdown and an increase in consumption in chronic/excessive consumers, who tend to be adults [18]. Among young adults aged 18–24, 27.4% reported having started or increased substance use to cope with stress or emotions related to the COVID-19 pandemic [19].

This study aims to observe and critically analyze the results of toxicological examinations on drivers performed under police mandate, in order to understand the influence of the pandemic on alcohol and drug consumption among drivers.

## 2. Materials and Methods

### 2.1. Population and Context

The authors consecutively collected the results of toxicological analyses on blood samples required by the Emergency Department of the University Hospital Policlinico di Bari (about 1000 beds) in southern Italy. The analyses were carried out at the Forensic Toxicology Laboratory of the Institute of Forensic Medicine of the Consortium University Hospital of the Bari Polyclinic.

The study period is between 1 January 2018 and 31 December 2021. The blood samples were taken from drivers due to police mandate.

### 2.2. Measurements

After the pseudo-anonymization of the data, the following variables were collected:

- Gender;
- Age;
- Reason for the request;
- Date of the analysis;
- Result of the toxicological analyses (primary outcome).

The extracted sample was then further divided according to the access date.

Specifically, the accesses were divided according to the period before and after 8 March 2020, the day on which the measures were implemented of the decree-law 23 February 2020, n.6, containing urgent measures regarding containment and management of the epidemiological emergency from COVID-191.

The sample thus defined was further subdivided into two subsamples based on the access date (pre-COVID/COVID) comprising 337 observations relating to the "pre-COVID" period (1 January 2018–8 March 2020) and 257 observations relating to the "COVID-19" period (between 9 March 2020 and 31 December 2021).

### 2.3. Toxicological Investigations

To carry out the chemical–toxicological investigations, the personnel of the Toxicology Laboratory took 30 cm$^3$ of blood in tubes containing EDTA, on which the following toxicological investigations were carried out.

2.3.1. Alcohol

The sample (1 mL of blood), after adding isopropyl alcohol, was analyzed in GC/HS gas chromatography under the conditions shown in Table 1.

**Table 1.** Parameter and setting/type of GC/HS for alcohol detection.

| Parameter | Setting or Type |
| --- | --- |
| Gas chromatograph | 7890A FID Agilent Technologies |
| HS detector | 7694E Agilent Technologies |
| Column | DB-ALC2 Agilent Technologies (30 m, I.D.0.32, 1.20 μm film) |
| Flows | Nitrogen 0.6 Atm; hydrogen 0.8 Atm; air 1.2 Atm |
| Injector temperature | 150 °C; |
| Column temperature | isotherm of 40 °C for 4 min; gradient 40–80 °C at 10 °C/min; gradient 80–120 °C at 25 °C/min |
| Detector temperature | 150 °C |
| Detector sensitivity | 8% (with 6 mcl) |
| Recording speed | 2 cm/min |
| Standard | 0.5 g/L ethanol solution |

This analysis allows us to highlight the concentration of ethyl alcohol present in the blood, as well as the dissolved gases and volatile solvents with pharmacological and toxic effects, such as ether and chloroform.

Since the study aimed at analyzing the variation in the consumption of alcohol or drugs in a broad context during the pandemic period, we considered the threshold value indicated by the highway code, which identifies the alcohol value >0.05 g/dL as not up to standard for those who have been licensed for at least three years, and equal to 0.00 g/dL for new drivers.

Because of this, we further subdivided the sample according to age, considering the reference alcohol value equal to 0 g/dL for subjects up to 21 years of age and 0.5 g/dL for those over the age of 21.

2.3.2. Drugs of Interest

The "Triage®8 Panel for Drugs of Abuse Test Kit" was used as a screening method to verify the presence of humoral markers attributable to drugs.

The Triage-8 is a rapid immunochemical test that allows us to verify the presence of the analytes on a blood matrix shown in Table 2.

**Table 2.** Drug screened with Triage®8 Panel.

| Abbreviation | Substance | Cut-Off |
| --- | --- | --- |
| AMP | Amphetamines | 500 ng/mL |
| mAMP | Methamphetamine | 500 ng/mL |
| BAR | Barbiturates | 200 ng/mL |
| BZO | Benzodiazepines | 200 ng/mL |
| COC | Cocaine | 150 ng/mL |
| EDDP | Methadone metabolite | 100 ng/mL |
| OPI | Opiates | 300 ng/mL |
| THC | Cannabinoids | 50 ng/mL |
| TCA | Tricyclic Antidepressants | 1000 ng/mL |

In cases in which the qualitative test results were positive for a substance, the quantitative determination of the same on a blood matrix was carried out by means of mass gas chromatography (GC/MS).

The instrumental analysis was carried out in Total Ion (mass range 40–500) on the basis of the mass spectra of the highlighted peaks. This analysis allows the identification of the substances present, both in relation to the standard mixture with a known title and by comparison with the spectra of over 70,000 substances of toxicological interest present in n. 3 libraries (data Station: "GC/MS Forensic Toxicological Database", "Hazardous Substances Data Bank", "NIST98").

The final extracts were analyzed in Mass Spectrometry with the instrumentation shown in Table 3.

**Table 3.** Parameter and setting/type of instrumentation for drug detection.

| Parameter | Setting or Type |
|---|---|
| Solvents | |
| | RP-ACS (Baker), HPLC grade (Carlo Erba) |
| Equipment | |
| Precision balance | E. Mettler, with 4 decimal digits |
| Centrifuge | Carlo Erba |
| Digital scale | Kern |
| Solid phase extraction equipment | Varian |
| Bond-Elut Certify columns | Varian |
| Toxi-Lab Tubes A and B | Varian, The Netherlands |
| Columns | Extrelut® NT3 (Merck, Germany). |
| Gas chromatograph with mass detector | |
| Gas chromatograph | Hewlett-Packard 6890 Plus |
| Mass detector | HP 5973N (Agilent Technologies, Milan, Italy) |
| Capillary column | HP 5-MS fused-silica (30 m × 0.32 mm; film thickness 0.30 μm) |
| Injector temperature | 250 °C |
| Column temperature, programmed | at 100 °C for 2 min; from 100 to 180 °C at 40 °C/min; from 180 to 290 °C at 10 ° C per minute; at 290 for 6 min. |
| Solvent delay | 3 min |
| Quantity analyzed | 1–2 μL |

### 2.4. Statistical Analysis

The data were analyzed using JAMOVI-Electron statistical software.

The quantitative variables were expressed as the mean, and the distribution normality was verified by the joint interpretation of Q–Q plots, skewness, kurtosis, and the Shapiro–Wilk test. The qualitative variables were expressed in proportion. Quantitative variables were compared using the t-test and the Levene test when parametric, while the Wilcoxon-Mann–Whitney test was used for non-parametric variables. The distribution of qualitative variables was evaluated using the $\chi^2$-test and the Fisher's Exact Test if the expected frequencies were <5. The Odds Ratio (OR) was used as a measure of association and calculated using the logistic regression. The variances were compared using the ANOVA test. Results with $p < 0.05$ were considered significant.

### 3. Results

In the period between 1 January 2018 and 31 December 2021, 684 samples were taken at the Forensic Toxicology Laboratory of the University of Bari. The number of samples totaled 594, including 337 from the pre-COVID-19 period and 257 from the COVID-19 period. The observations excluded were those relating to the assessment of blood alcohol levels in subjects in whom a liver transplant was planned or in subjects treated with specific chemotherapy drugs which, according to guidelines, provide for the monitoring of blood alcohol levels. The general and demographic characteristics of the sample are summarized in Table 4.

**Table 4.** Demographic information. * *p*-value < 0.05.

|  | Pre-COVID-19 (%) | COVID-19 (%) | Total |
|---|---|---|---|
| **Genre** |  |  |  |
| Male | 281 (58.3) | 201 (41.7) | 482 |
| Female | 56 (51.9) | 52 (48.1) | 108 |
| Not indicated | 0 (0.0) | 4 (100.0) | 4 |
| **Age** |  |  |  |
| <21 y.o. | 53 (62.4) | 32 (37.6) | 85 |
| ≥21 y.o. | 284 (55.9) | 224 (44.1) | 508 |
| **Mean (y.o.)** | 37.8 * (96.7) | 40.7 * (3.3) | 39.1 |
| Median | 35.0 (94.6) | 39.0 (5.4) | 37 |
| Standard Deviation | 15.9 (98.1) | 16.4 (1.9) | 16.2 |
| Range | 14–86 | 16–85 | 14–86 |

The age distribution in the sample was normally distributed, and the average age was significantly higher in the COVID group [Figure 1].

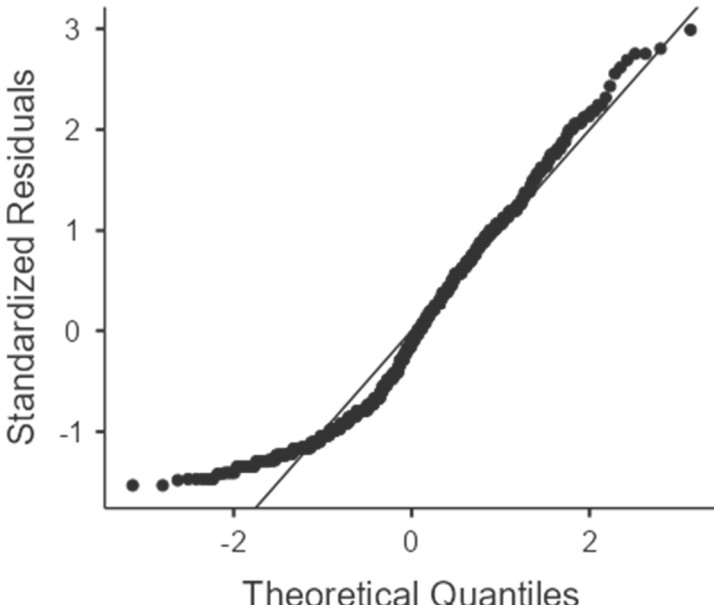

**Figure 1.** Q–Q age plot.

Considering all the observations, the qualitative analysis of the blood concentration of substances showed 123 (20.7%) subjects with a blood alcohol level higher than 0.5 g/L, 21 (3.9%) subjects positive for cocaine, and 29 (5.4%) cannabis-positive subjects. A total of 26.3% (156) of the subjects were found to be unfit to drive due to the intake of one or more substances. The positivity was not detected for the other analytes evaluated. There were 16 subjects who tested positive for several substances (one positive for alcohol, cocaine, and cannabis; four positive for cocaine and cannabis; three positive for alcohol and

cocaine; eight positive for alcohol and cannabis). The frequency of positivity to substances is represented by Table 5.

**Table 5.** Driving fitness. * *p*-value < 0.05.

| | Pre-COVID-19 | COVID-19 | Total |
|---|---|---|---|
| **Frequency of observations** | | | |
| Number of days | 797 | 692 | 1489 |
| Total observations | 337 | 257 | 594 |
| Observations-100 day | 42.28 | 37.14 | 39.89 |
| **Driving fitness** | | | |
| Alcohol >0.5 g/dL | 63 | 60 | 123 |
| Positive for cocaine | 13 | 8 | 21 |
| Positive for cannabis | **22 *** | **7 *** | 29 |
| Positive for other substance | 0 | 0 | 0 |
| Not suitable for driving | 86 | 70 | 156 |
| **Incidence rate (positive observations/observations-100 days)** | | | |
| Alcohol >0.5 g/dL | 1.49 | 1.62 | |
| Positive for cocaine | 0.31 | 0.22 | |
| Positive for cannabis | **0.52 *** | **0.19 *** | |
| Positive for other substance | 0.00 | 0.00 | |
| Not suitable for driving | 2.03 | 1.88 | |

Among those unfit to drive, the average alcohol level was 1.56 g/L, with a normal distribution. In the COVID-19 period, the average alcohol level was significantly higher (1.73 g/L in the COVID-19 period, 1.40 g/L in the pre-COVID period, *p* < 0.05). The alcohol level measured was directly proportional to age (standard β 0.3, confidence interval 0.11–0.48). In the group of newly licensed patients (<21 y.o.) the mean alcohol level was significantly lower (new drivers 0.97 g/L, non-new drivers 1.62 g/L, *p* < 0.05).

The determinants of substance use and fitness to drive are analyzed in Table 6. Cannabis use was found to be a positive predictor of cocaine use. Young people were more often cannabis-positive. As age increases, there is a greater likelihood of being fit to drive [Table 6].

**Table 6.** Determinants of substance consumption. Only significant results.

| | Alcohol over the Limits | Positive for Cocaine | Positive for Cannabis | Suitable for Driving |
|---|---|---|---|---|
| | **Odds Ratio (I.C. 95%)** | | | |
| **Age** | / | / | 0.97 (0.94–0.99) | 1.01 (1.00–1.03) |
| **Alcohol over the limits** | / | / | / | / |
| **Positive for cocaine** | / | / | 4.57 (1.36–15.44) | / |
| **Positive for cannabis** | / | 4.62 (1.37–15.56) | / | / |
| **COVID-19 period** | / | / | / | / |
| **Male gender** | / | / | / | / |

## 4. Discussion

This study showed that just over a quarter (26.3%) of the subjects subjected to control were unsuitable for driving due to the intake of one or more substances. The pandemic period did not lead to a substantial change in the prevalence of alcohol and drug use in the sample under examination. Younger subjects (<21 years) were more frequently found to be unfit to drive due to substance intake.

The percentage of subjects analyzed with a blood alcohol level above the legal threshold (blood alcohol level >0.5 g/dL) was 20.7%. This figure is higher than what emerged from the Italian epidemiological study "PASSI", relating to the three-year period 2017–2020, in which only 6.5% of the interviewees declared that they were driving under the influence of alcohol [20]. In our dataset, no significant differences were found in the two groups (pre- and post-lockdown). These results must also be interpreted in relation to the reduced number of cars circulating during the lockdown period and the increase in checks compared to the previous period. Since the data on the actual number of cars in circulation are not available, it is not possible to comment on a variation, in absolute terms, of traffic offenses in the two historical periods. Therefore, although the data show that alcohol consumption in the sample under examination has apparently remained unchanged, it is reasonable to think that, given the fewer drivers on the road due to the restrictive measures, there has been a relative increase in alcohol use in the general population. This accords with literature data indicating an increase in alcohol consumption in the at-risk population in the rest of the world during the lockdown [16]. In Italy, on the other hand, studies suggest that there has been a decrease in the consumption of alcoholic beverages in the general population, but an increase in populations at greater risk [18,20].

In relation to the entire period examined, the average alcohol level was significantly lower in young subjects (<21 years, 0.97 g/L) than in older drivers (1.62 g/L). The data are consistent with similar studies [21]. These data could also be related to the difference between the threshold for the quantity of alcohol consumed as a limit value between the two groups of drivers (0.0 mg/dL of under 21 s vs. <0.5 mg/dL of the rest of the population). It is known, however, that during the lockdown and following the measures to restrict social contacts, many "very young" people have consolidated at-risk habits such as "binge drinking", the so-called "drinking to get drunk", with an intake of a high quantity of alcoholic drinks on one occasion [22].

The results showed an increase in blood alcohol levels during the pandemic among people with alcohol levels higher than the values allowed by the law (1.73 g/L vs. 1.40 g/L). These data can be interpreted as an indicator of the tendency of alcohol users to increase alcohol intake during the period of social isolation, in line with the results of Alladio et al. [18]. In other words, the data collected in this study allow us to highlight an increase in alcohol consumption in that segment of the population already prone to its consumption, probably as a defensive mechanism against the stress generated by social isolation and the reduction of contacts.

In any case, the increased alcohol level for those who already consumed alcohol remains a figure that emerged only for drivers subjected to road checks. The increase in online alcohol sales does not contradict what emerged from our study: in fact, due to the restrictions, those who could move were exclusively individuals with proven health reasons or work needs. In other words, these data only indicate an increase in the quantity of alcohol introduced by those drivers who regularly, even in the past, drank alcohol.

As regards the use of cocaine, the percentage of positivity found by the authors remained unchanged in the two periods under examination, with an average overall positivity of 3.9%. The data on cocaine use while driving are in line with other studies, performed in pre-pandemic periods, on populations of drivers selected for suspected cocaine use (5% in Augsburger et al.) [23]. Similarly, research aimed at analyzing the use of cocaine through its presence in wastewater confirms that this consumption has remained unchanged [24].

Regarding the use of cannabis and its derivatives, significantly lower consumption is reported compared to what emerged from other studies (5.4% vs. 59% by Augsburger et al.) [23]. In the sample under study, the use of cannabinoids seems to be associated with a lower age of the sample (under 21 years old). These data are in agreement with the data already present in the literature, which suggests that age is a fundamental risk factor for the intake of these substances, with an increase in use in the age group of the baby boomer generation [25].

With regard to the simultaneous intake of multiple substances, it is interesting to note that, from our data, the consumption of cannabis is positively associated with the simultaneous use of cocaine. In fact, the co-use of cocaine and cannabis is well described in the literature [26].

Instead, in the sample under examination, alcohol consumption is not statistically related to the use of other abusive substances.

The main limitation of the study is that the authors only observed a single center, which further reduces the representativeness of the sample. Finally, in the absence of exact data on the vehicles circulating during the different periods under examination, it was not possible to calculate the positivity rates in toxicological tests and its variation over time.

## 5. Conclusions

The lockdown has led to an increase in stress, which might stimulate the increase in or starting of substance use as a way of coping. The present study showed that the drivers stopped by the police during the pandemic were less likely to be positive for cannabis compared with the pre-COVID population. The blood levels of alcohol among drivers with alcohol levels above the legal threshold for driving were higher during the pandemic. However, these results might be linked to the age of the stopped drivers, which was significantly higher during the pandemic period. Age was found to be negatively associated with cannabis positivity. More generally, older drivers were more likely to be suitable for driving according to Italian law. Finally, cannabis positivity was a determinant of cocaine positivity in the sample.

The study did not show an increase in the incidence of alcohol and drug positivity among drivers. However, as the drivers sampled during the pandemic were older than before, this study might not be representative of the effect of the pandemic and the lockdown on young people's health. Whether and how young people suffered from psychological disorders because of pandemic period might be the object of further study.

**Author Contributions:** Conceptualization, M.M. and M.B.; methodology, G.S. and L.A.; validation, M.M. and M.B.; investigation, G.S. and L.A.; data curation, M.B. and F.P.; writing—original draft preparation, F.P., A.S., S.N., L.B., B.P.D.L. and M.B.; writing—review and editing, M.M., B.S., F.P. and M.B.; supervision, M.M. and M.B. All authors have read and agreed to the published version of the manuscript.

**Funding:** This research received no external funding.

**Institutional Review Board Statement:** All patients provided informed consent, allowing the use of anonymous clinical information to be used for research purposes. The study was carried out in accordance with the Declaration of Helsinki 1995 (as revised in Edinburgh 2000) and was approved by the Institutional Review Board of Comitato Etico Indipendente, Azienda Ospedaliero-Universitaria and the protocol number is 0023264/09/03/2023/AOUCPG23/COMET/P.

**Informed Consent Statement:** Written informed consent has been obtained from the patient(s) to publish this paper.

**Data Availability Statement:** The data presented in this study are available on request from the corresponding author. The data are not publicly available due to privacy policy.

**Conflicts of Interest:** The authors declare no conflict of interest.

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
