# Peer review of "Alcohol and Drug Consumption among Drivers before and during the COVID-19 Pandemic: An Observational Study"

_ejihpe, doi:10.3390/ejihpe13050068_

Round 1

Reviewer 1 Report

This manuscript is quite confusing in terms of its aims and methods and seems to overstep its implications in the discussion.  

In many places the authors state facts about characteristics of individuals who may use alcohol or different substances, but these are not referenced.  Since no data on the psychological status of the subjects in the study were collected, one cannot draw any inferences about why individuals were using substances.  (lines 83-84 'alcohol was the most used substance...in order to reduce feelings of anxiety and/or stress...") line 97: "as for substance use disorders they mainly concerned women, university students and racial and ethnic minorities." No citations and again, there was no measure of substance use disorders in the data collected. Line 260: :..many young people have consolidated habits at risk such as binge drinking.." no reference. Line 266 cites a study that found psychologically fragile subjects increasing alcohol as a defense mechanism against stress but there there are no psychological measures in this study. 

In the beginning (lines 65-74) a lot of detail is provided about different types of restrictions during the Covid-19 public health emergency, but since these aren't related to the analyses, they are unnecessary.  The authors pose only a pre-post analysis around the date of initial implementation of restrictive measures. 

In the paragraph starting on line 103, a lot of information is provided about police reports of drug and alcohol related accidents, but the dates are confusing and its not clear if these rates are from toxicology reports done by the police.  

There is no concise research question posed, or sub-questions.  It is implied that police reports suggested increases in 2020 in substance related accidents, but the time frame of this study doesn't line up directly. Also, there is not discussion as to why verification at an ED is an important contribution to our understanding, especially since information was not presumably collected on issues such as psychological state, stress, and substance disorders (certainly possible someone who has a substance related accident does not have a substance disorder). Further, some accidents could be minor yet still involve substance use and therefore not involve an ED visit for drivers/passengers.  How does that relate to the population that needs medical care from a substance related accident? How are those who caused the accident distinguished from those injured in an accident who might also be using substances but did not cause the accident?

The methods, while detailed on the chemistry, lack information about the geographic location of the ED where blood samples were taken, any relevant discussion of how it operated and how/when blood samples were drawn, if it's a 100% sample of all ED patients involved in a vehicle accident in this region during the time frame, whether consent was obtained to use the samples for research purposes or if use of such data went before an ethics review committee.  How do we know only drivers were sampled? Could there also be injured passengers?  

Some other small issues with understanding what is being presented:  Line 105, what year is being referred to? Line 110, what do the numbers mean in the parentheses? Line 115, what is the number '17'? Line 133, is the word supposed to be "gender"?  In Table V, why are days entered?  Table VI, the analysis should be explained, and entering the missing data would be helpful.

Finally, line 312 implies that actual rates of substance use went up during Covid restrictions (even though the data show decreases in the toxicology rates for all substances reported) because less vehicles on the road meant the same proportions in the sampled ED patients actually underrepresented rates because there was less driving.  This seems to be the main point of the study. However, the study is comparing those coming to the ED for vehicle accidents involving substances both before and after Covid restrictions, and the proportion of positive substance toxicology was lower.  So I don't see that the conclusions drawn are correct.

Author Response

REVIEWER 1

  1. This manuscript is quite confusing in terms of its aims and methods and seems to overstep its implications in the discussion.
  2. In many places the authors state facts about characteristics of individuals who may use alcohol or different substances, but these are not referenced.  Since no data on the psychological status of the subjects in the study were collected, one cannot draw any inferences about why individuals were using substances.  (lines 83-84 'alcohol was the most used substance...in order to reduce feelings of anxiety and/or stress...") line 97: "as for substance use disorders they mainly concerned women, university students and racial and ethnic minorities." No citations and again, there was no measure of substance use disorders in the data collected. Line 260: :..many young people have consolidated habits at risk such as binge drinking.." no reference. Line 266 cites a study that found psychologically fragile subjects increasing alcohol as a defense mechanism against stress but there there are no psychological measures in this study. 
  3. In the beginning (lines 65-74) a lot of detail is provided about different types of restrictions during the Covid-19 public health emergency, but since these aren't related to the analyses, they are unnecessary.  The authors pose only a pre-post analysis around the date of initial implementation of restrictive measures. 
  4. In the paragraph starting on line 103, a lot of information is provided about police reports of drug and alcohol related accidents, but the dates are confusing and it’s not clear if these rates are from toxicology reports done by the police.  
  5. There is no concise research question posed, or sub-questions.  It is implied that police reports suggested increases in 2020 in substance related accidents, but the time frame of this study doesn't line up directly. Also, there is not discussion as to why verification at an ED is an important contribution to our understanding, especially since information was not presumably collected on issues such as psychological state, stress, and substance disorders (certainly possible someone who has a substance related accident does not have a substance disorder). Further, some accidents could be minor yet still involve substance use and therefore not involve an ED visit for drivers/passengers.  How does that relate to the population that needs medical care from a substance related accident? How are those who caused the accident distinguished from those injured in an accident who might also be using substances but did not cause the accident?
  6. The methods, while detailed on the chemistry, lack information about the geographic location of the ED where blood samples were taken, any relevant discussion of how it operated and how/when blood samples were drawn, if it's a 100% sample of all ED patients involved in a vehicle accident in this region during the time frame, whether consent was obtained to use the samples for research purposes or if use of such data went before an ethics review committee.  How do we know only drivers were sampled? Could there also be injured passengers?  
  7. Some other small issues with understanding what is being presented:  Line 105, what year is being referred to? Line 110, what do the numbers mean in the parentheses? Line 115, what is the number '17'? Line 133, is the word supposed to be "gender"?  In Table V, why are days entered?  Table VI, the analysis should be explained, and entering the missing data would be helpful.
  8. Finally, line 312 implies that actual rates of substance use went up during Covid restrictions (even though the data show decreases in the toxicology rates for all substances reported) because less vehicles on the road meant the same proportions in the sampled ED patients actually underrepresented rates because there was less driving.  This seems to be the main point of the study. However, the study is comparing those coming to the ED for vehicle accidents involving substances both before and after Covid restrictions, and the proportion of positive substance toxicology was So I don't see that the conclusions drawn are correct.

RESPONSE

  1. we adjusted the aims according to your suggestions, trying to make it clearer.
  2. We corrected the issue by eliminate unreferenced sentences or adding citations in the right place.
  3. we eliminate superfluous information in the introduction.
  4. we eliminated the useless informations.
  5. the aim of the present research was to describe and critically analyse the results of toxicological examines performed in the ED due to a police mandate. Hence, all the tested subjects were drivers. We clarified the aims according to your suggestion.
  6. as stated in the methods the study was set in Bari, southern Italy. The samples were taken from drivers stopped by the police. We only used anonymized data and we had the Institutional review Board Statement (lines 344-348)
  7. We have corrected the introduction by removing some unnecessary information and so we changed lines 103-… In table V days are entered in order to give a 100-day incidence of positivity to substances. In Table VI there are the results of the regression analysis, so we entered the Odds ratio and Confidence Interval of significant associations while the missing data are insignificant. We adjusted the methods to improve the understanding of statistical analysis.
  8. we rewrote the conclusion according to your suggestion.

Reviewer 2 Report

For the abstract, please include the dates of “before” and “after the lockdown measures”.

The English needs some work.

The information about road accidents in the introduction should be in the methods, unless they should that the information has limitations that the current study will address.

They need to show that there was not a change in testing of drivers.

They should provide a context of why the information of Bari is of value.

What is “Genre”?

“Drugs of abuse” is stigmatizing. Please consider alternatives, for example, “drugs of interest”.

“t-Student” should be “t-tests”

For chi-square, did the investigators mean “Fisher’s Exact Test” or “Yates continuity correction”?

The first sentence of results is not understandable. Please revise.

For Table 4, please include percentages.

Please revise the statement “As you get older…”. The manuscript is being considered for publication in a professional journal.

“Both would find comfort” sentence needs to be revised.

There was no change between the periods examined. Therefore, the authors should not speculate about fragile people or defense mechanisms.

There were multiple comparisons. Please designate primary outcome and then correct for multiple comparisons for other comparisons.

The conclusions are stigmatizing and not well supported.

Author Response

REVIEWER 2

  1. For the abstract, please include the dates of “before” and “after the lockdown measures”.
  2. The English needs some work.
  3. The information about road accidents in the introduction should be in the methods, unless they should that the information has limitations that the current study will address. They need to show that there was not a change in testing of drivers.
  4. They should provide a context of why the information of Bari is of value.
  5. What is “Genre”?
  6. “Drugs of abuse” is stigmatizing. Please consider alternatives, for example, “drugs of interest”.
  7. “t-Student” should be “t-tests”
  8. For chi-square, did the investigators mean “Fisher’s Exact Test” or “Yates continuity correction”?
  9. The first sentence of results is not understandable. Please revise.
  10. For Table 4, please include percentages.
  11. Please revise the statement “As you get older…”. The manuscript is being considered for publication in a professional journal.
  12. “Both would find comfort” sentence needs to be revised.
  13. There was no change between the periods examined. Therefore, the authors should not speculate about fragile people or defense mechanisms.
  14. There were multiple comparisons. Please designate primary outcome and then correct for multiple comparisons for other comparisons.
  15. The conclusions are stigmatizing and not well supported.

RESPONSE

  1. We included the dates of “before” (1 Jan 2018 to 9 Mar 2020) and “after the lockdown measures” (9 Mar 2020 to 31 Dec 2021).
  2. We improved the language.
  3. We changed the introduction by modifying the information concerning car accident.
  4. We adjusted the aims trying to make it cleare.
  5. Sorry, it was “gender”.
  6. We wrote “drugs of interest”.
  7. We wrote “t-tests”
  8. We used the “Fisher’s Exact Test” when the expected frequencies were <5. We corrected the method trying to better explain the statistical analysis.
  9. We changed “The total accesses to the Forensic Toxicology Laboratory of the University of Bari in the period between 1.1.2018 and 31.12.2021 were equal to 684” with “In the period between 1 Jan 2018 and 31 Dec 2021, 684 samples were taken at the Forensic Toxicology Laboratory of the University of Bari.”
  10. We adjusted the table 4 according to your suggestions
  11. We changed “As you get older, you are more likely to be fit to drive” in “As age increases, there is a greater likelihood of being fit to drive.”
  12. We changed “Both would find comfort in the trend of alcohol consumption in the population at risk in the rest of the world that attests an increase during the lockdown” with “This accords with literature data indicating an increase in alcohol consumption in the at-risk population in the rest of the world during the lockdown.”
  13. We avoided unsupported speculations by correcting the entire manuscript.
  14. The primary outcome is the result of the toxicological analysis, as we clearly stated by modifying the methods.
  15. we rewrote the conclusion according to your suggestions

Reviewer 3 Report

This is a very interesting study in which the authors do not take best advantage of the data available to them. They failed to present both rates and numbers for each of the parameters being measured. They presented  narrative conclusions that were almost certainly correct, but not evident from the data presented.

This is not a “cross-sectional” study. It is a pre/post exposure study since it presents data from two different time periods.

Line 20 of the abstract references “a growing trend.” This is not correct. The data suggest an influence of the pandemic closure on substance abuse and driving behavior.

While the authors have no access to data relative to numbers of cars on the road or Kilometers driven, they should have easy access to gasoline tax revenues by month – which could be used as a surrogate for these parameters, so the numbers of observations can be presented as “per million liters of gasoline consumed, or some similar measure.  Anticipating that driving was reduced 50% to 80% during the pandemic, and anticipating that almost all gasoline was used for automobile travel, such an adjustment is of major importance.

Since there were fewer days of observation, a comparison should be traffic stops per 100 days of observation, with re-calculation of p values as appropriate.

Authors make the point that patterns of substance use observed vary by whether the driver was over or under 21 years of age. This being the case, subsequent data should be reported separately for each age group, and, if appropriate, by both age and gender.

Making these adjustments will almost certainly create the impression that such observations were more frequent during the pandemic period and more boldly present the findings on alcohol use by those over 21 and use of cannabis by those under 21.

Once these adjustments are made, with full presentation of the data, the “limitations” paragraph can be re-written to simply reflect that, since this is an observational study from a single center, these results might not reflect the experience in other jurisdictions.

Author Response

REVIEWER 3

  1. This is a very interesting study in which the authors do not take best advantage of the data available to them. They failed to present both rates and numbers for each of the parameters being measured. They presented narrative conclusions that were almost certainly correct, but not evident from the data presented.
  2. This is not a “cross-sectional” study. It is a pre/post exposure study since it presents data from two different time periods
  3. Line 20 of the abstract references “a growing trend.” This is not correct. The data suggest an influence of the pandemic closure on substance abuse and driving behavior.
  4. While the authors have no access to data relative to numbers of cars on the road or Kilometers driven, they should have easy access to gasoline tax revenues by month – which could be used as a surrogate for these parameters, so the numbers of observations can be presented as “per million liters of gasoline consumed, or some similar measure.  Anticipating that driving was reduced 50% to 80% during the pandemic, and anticipating that almost all gasoline was used for automobile travel, such an adjustment is of major importance. Since there were fewer days of observation, a comparison should be traffic stops per 100 days of observation, with re-calculation of p values as appropriate. Authors make the point that patterns of substance use observed vary by whether the driver was over or under 21 years of age. This being the case, subsequent data should be reported separately for each age group, and, if appropriate, by both age and gender.
  5. Making these adjustments will almost certainly create the impression that such observations were more frequent during the pandemic period and more boldly present the findings on alcohol use by those over 21 and use of cannabis by those under 21.
  6. Once these adjustments are made, with full presentation of the data, the “limitations” paragraph can be re-written to simply reflect that, since this is an observational study from a single center, these results might not reflect the experience in other jurisdictions.

RESPONSE

  1. we rewrote the conclusion according to your suggestions
  2. We modified the paper by avoiding the expression “cross-sectional
  3. Thank you for correcting us, we change according to your suggestions.
  4. Thank you for this interesting suggestion. We considered the reduction in fuel consumption in Italy during the pandemic as a surrogate the number of circulating cars. We also referred to some direct measures performed by Italian agencies on car circulation during pandemic.
  5. We compare the positive observations pre- and post- COVID by referring to the 100 days- incidence rate, as you suggested
  6. We adjusted the “Limitation” paragraph following your suggestions.

Round 2

Reviewer 1 Report

The authors have made a few changes and substantially rewritten the conclusion to this study which showed essentially no increase in alcohol or drug use among drivers stopped and mandated to have toxicology tests in this one hospital   pre and post covid restrictions.  This will be a useful study to take in context since others have reported increases in substance use due to covid.  It may however mean that because of travel restrictions and fewer people driving, it did not show up in a driver only cohort.  Also interesting is that people under 21 were less represented in the post covid  sample, probably meaning they had less reason to drive due to closed educational institutions and/or less engagement with allowed employment exemptions. 

Author Response

  1. The authors have made a few changes and substantially rewritten the conclusion to this study which showed essentially no increase in alcohol or drug use among drivers stopped and mandated to have toxicology tests in this one hospital   pre and post covid restrictions.  This will be a useful study to take in context since others have reported increases in substance use due to covid.  It may however mean that because of travel restrictions and fewer people driving, it did not show up in a driver only cohort.  Also interesting is that people under 21 were less represented in the post covid  sample, probably meaning they had less reason to drive due to closed educational institutions and/or less engagement with allowed employment exemptions.

RESPONSE

  1. Following to the reviewers’ comments we tried to improve the quality of the manuscript. Thank you.

Reviewer 2 Report

Please change to Fisher's Exact Test in the statistical section

Author Response

REVIEWER 2

  1. Please change to Fisher's Exact Test in the statistical section

RESPONSE 2

  1. We modified the methods according to your suggestion.
